# Systemic Inflammation and the Breakdown of Intestinal Homeostasis Are Key Events in Chronic Spinal Cord Injury Patients

**DOI:** 10.3390/ijms22020744

**Published:** 2021-01-13

**Authors:** David Diaz, Elisa Lopez-Dolado, Sergio Haro, Jorge Monserrat, Carlos Martinez-Alonso, Dimitrios Balomeros, Agustín Albillos, Melchor Alvarez-Mon

**Affiliations:** 1Department of Medicine, University of Alcalá, Crta N-II km 33.6, Alcalá de Henares, 28871 Madrid, Spain; david.diaz@uah.es (D.D.); lamidolado@gmail.com (E.L.-D.); sergioharogiron@gmail.com (S.H.); Jorge.monserrat@uah.es (J.M.); Agustin.albillos@uah.es (A.A.); 2Institute Ramón y Cajal for Health Research (IRYCIS), Ctra. Colmenar Viejo, 28034 Madrid, Spain; 3Biomedical Institute for Liver and Gut Diseases (CIBEREHD), Instituto de Salud Carlos III, Av. Monforte de Lemos, 3-5, 28029 Madrid, Spain; 4Service of Rehabilitation, National Hospital for Paraplegic Patients, Carr. de la Peraleda, S/N, 45004 Toledo, Spain; 5Department of Immunology and Oncology, Universidad Autónoma de Madrid, Calle Darwin, 3, 28049 Madrid, Spain; cmartineza@cnb.csic.es (C.M.-A.); dbalomenos@cnb.csic.es (D.B.); 6Service of Gastroenterology, University Hospital Ramón y Cajal, 28034 Madrid, Spain; 7Immune System Diseases and Oncology Service, University Hospital “Príncipe de Asturias”, Crta N-II km 33.6, Alcalá de Henares, 28871 Madrid, Spain

**Keywords:** monocyte, cytokines, chronic spinal cord injury, bacterial translocation, gut barrier damage, systemic inflammation

## Abstract

Our aim was to investigate the subset distribution and function of circulating monocytes, proinflammatory cytokine levels, gut barrier damage, and bacterial translocation in chronic spinal cord injury (SCI) patients. Thus, 56 SCI patients and 28 healthy donors were studied. The levels of circulating CD14^+high^CD16^−^, CD14^+high^CD16^+^, and CD14^+low^CD16^+^ monocytes, membrane TLR2, TLR4, and TLR9, phagocytic activity, ROS generation, and intracytoplasmic TNF-α, IL-1, IL-6, and IL-10 after lipopolysaccharide (LPS) stimulation were analyzed by polychromatic flow cytometry. Serum TNF-α, IL-1, IL-6 and IL-10 levels were measured by Luminex and LPS-binding protein (LBP), intestinal fatty acid-binding protein (I-FABP) and zonulin by ELISA. SCI patients had normal monocyte counts and subset distribution. CD14^+high^CD16^−^ and CD14^+high^CD16^+^ monocytes exhibited decreased TLR4, normal TLR2 and increased TLR9 expression. CD14^+high^CD16^−^ monocytes had increased LPS-induced TNF-α but normal IL-1, IL-6, and IL-10 production. Monocytes exhibited defective phagocytosis but normal ROS production. Patients had enhanced serum TNF-α and IL-6 levels, normal IL-1 and IL-10 levels, and increased circulating LBP, I-FABP, and zonulin levels. Chronic SCI patients displayed impaired circulating monocyte function. These patients exhibited a systemic proinflammatory state characterized by enhanced serum TNF-α and IL-6 levels. These patients also had increased bacterial translocation and gut barrier damage.

## 1. Introduction

Spinal cord injury (SCI) is a cause of severe health problems and disability [1]. The acute stage of the disease is characterized by the induction of a neurological lesion of the spinal cord and the associated clinical manifestations and systemic stress responses determined by the etiopathogenesis of the SCI [2,3,4,5,6,7]. The management of acute SCI has improved in recent years, with a dramatic reduction in mortality [8,9]. However, patients with chronic SCI suffer a high incidence of medical complications, such as infections, metabolic diseases, cardiovascular events, and recurrent episodes of impaired general health [10,11].

It is well known that acute SCI patients show an intense disturbance of immune-inflammatory responses, which includes hallmarks of inflammation and immunodeficiency [12,13]. At injury sites in the spinal cord, the infiltration of immune cells, including monocytes/macrophages, is involved in determining the extent of the initial tissue damage and causing secondary neural destruction during the first several weeks postinjury [2,14]. Furthermore, the secondary systemic immunodepression that is observed in patients with SCI has been associated with a predisposition to infection complications [15,16]. Acute inflammation after SCI also appears to be related to patient functional outcomes [17]. The mechanisms that contribute to acute SCI-associated immune disturbance appear to be multifactorial, including traumatic and surgery stress-related neuro-endocrine responses, infection complications, and central and autonomous nervous system alterations and lesions [18,19].

The function of the immune system in patients with chronic SCI remains partially characterized. Abnormal levels of several circulating cytokines have been described [20]. Decreased natural killer (NK) cell counts and cytotoxic activity levels have been reported, and T lymphocytes from patients with chronic SCI also show abnormal function [6,21]. Furthermore, inflammatory cells have been detected in human spinal cord tissue years after the initial SCI [22].

Monocytes are bone marrow-derived cells that mediate essential regulatory and effector functions in innate and adaptative immunity [23]. Circulating human monocytes are phenotypically and functionally heterogeneous and are divided into three major subsets based on the expression of the lipopolysaccharide (LPS) receptor CD14 and the Fc*γ*RIII low-affinity IgG receptor CD16: classical (CD14^+high^CD16^−^), intermediate (CD14^+high^CD16^+^), and nonclassical (CD14^+low^CD16^+^) [23,24].

The recognition of microorganisms by proteins that recognize pathogen-associated molecular patterns, such as Toll-like receptors (TLRs), is critical for the activation of monocytes and development of the natural immune response [25]. TLR2, TLR4, and TLR9 recognize bacterial molecules, such as lipoteichoic acid, LPS, and unmethylated cytosine-phosphate-guanine DNA, respectively. Activated monocytes show relevant immunomodulatory activities, including the secretion of pivotal cytokines, such as the proinflammatory cytokines interleukin (IL)-6, IL-1 and tumor necrosis factor-alpha (TNF-α), and the anti-inflammatory cytokine IL-10 [26]. Monocytes are also important phagocytic cells [27]. Monocytes have been demonstrated to be involved in the pathogenesis of several organ-specific and systemic inflammatory diseases [28].

In experimental models of SCI, intestinal dysbiosis and increased gut permeability were recently demonstrated [29]. We propose that patients with chronic SCI suffer damage to the intestinal barrier, with secondary increased permeability that favors increased bacterial translocation, and a systemic inflammatory imbalance, with monocyte compromise. Intestinal fatty acid-binding protein (I-FABP) and zonulin are recognized protein markers of the integrity of the intestinal barrier. The hepatic synthesis of LPS-binding protein is promoted by LPS [30], and in several clinical settings, plasma LBP reflects long-term exposure to bacteria and their endotoxins [31,32].

We have focused our work on the study of circulating monocytes and the level of bacterial translocation and gut barrier damage in patients with chronic SCI. To avoid confounding factors, we focused our study on a homogeneous population of 56 chronic SCI patients without clinical infections or concomitant diseases with potential interactions with the immune system. In parallel, we studied 28 age- and sex-matched healthy controls (HCs). We analyzed the pattern of distribution of the CD14^+high^CD16^−^, CD14^+high^CD16^+^ and CD14^+low^CD16^+^ circulating monocyte subsets, as well as their TLR2, TLR4 and TLR9 expression levels. We also investigated the intracytoplasmic production of TNF-α, IL-1, IL-6 and IL-10 after LPS stimulation and the serum levels of these cytokines. Monocyte reactive oxygen species (ROS) production and phagocytic activity were also analyzed. We measured the serum levels of LBP, I-FABP, and zonulin to study bacterial translocation and gut barrier damage.

## 2. Results

### 2.1. Demographic Profile of Chronic SCI Patients

Table 1 shows the characteristics of the 56 chronic SCI patients and 28 HCs included in the analysis. No significant differences were found between chronic SCI patients and HCs with respect to age or sex distribution or the clinical and analytical variables studied.

At the inclusion time, 68% of the SCI patients were males and 32% women. Their mean age was 26.9286 ± 12.8797 years. The mean time of SCI onset was 11.6786 ± 9.0621 years, a time of evolution that certainly define them as chronic injuries. Regarding the SCI etiology, 54% were traumatic (traffic accidents, diving injuries and falls), and the rest of them non-traumatic, 5% birth-SCI doubt to labor dystocia, 11% inflammatory myelitis, 6% spinal vascular diseases, 11% spinal cord tumor sequelae, and the remaining 13% consequently to spina bifida.

Regarding the need for spinal surgery in the acute period of SCI, it was necessary in 76% of traumatic patients versus 56% of those whose injury was due to a non-traumatic aetiology. Traumatic patients mainly received vertebral fracture reduction and arthrodesis by plates or bars and screws. Only one of them required replacement of the osteosynthesis material a few weeks after the first surgery, due to loosening and incomplete reduction of the fracture. Non-traumatic patients were treated by laminectomies and drainage of abscesses or haematomas or, in the case of spina bifida, closure of the congenital spinal defect at birth. No statistically significant differences were found between both subgroups in this regard (Mann–Whitney Rank Sum Test, *p* = 0.134). Only 12% of the traumatic patients needed new spinal surgeries along their SCI time evolution, compared to 45% of the non-traumatic ones (Mann–Whitney Rank Sum Test, *p* = 0.001), mostly due to the need of an average of more than two spinal surgeries in the SB patients. In no case was any patient included in the study who had undergone surgery, spinal or any other type, in the last year. The average time since the last surgery in the present series was 3.6 years, being significantly longer in non-traumatic cases (Mann–Whitney Rank Sum Test, *p* = 0.016).

The neurological level of spinal damage was located within C1–C4, C5–C8, T1–T6, T7–T12 and lumbosacral metameras in 23.21%, 19.64%, 23.22%, 19.64% and 14.29% of the patients respectively, which implies that more than 66% of our patients showed SCI above T6, with higher expected signs and symptoms of autonomic disreflexia. With respect to the ASIA impairment score (AIS), 34% of the patients were AIS A, 21.43% AIS B, 19.64% AIS C, and 25% AIS D, meaning that although 66% of our patients showed incomplete lesions, only 44.64% showed motor incomplete injuries with different degrees of infralesional motor preservation and theoretically better mobility profiles. These results are in agreement with the quite good functional scores that these patients reached, both in the basic activities of daily living and in the ability to walk: the mean SCIM III score of our series was 60 ± 2.9818 over a maximum score of 100. When we analyzed the three SCIM III domains separately, we found acceptable performance in self-care (mean subscore = 14.5 ± 0.8886 over a maximum of 20), respiratory and sphincters management (mean score = 26.0714 ± 0.8816 over a maximum of 40) and mobility (mean subscore = 19.4464 ± 1.5289 over a maximum of 30). Locomotion was preserved in 32 patients, (mean WISCI II score of 12.1875 ± 1.1798 over a maximum value of 20), meaning of 57% of our patient series. It would be a quite surprising result, since we only had 44.64% motor incomplete injuries in our series and plus 66% of them showed SCI above T6, in which locomotion is less less likely to be preserved. A value of 12 in WISCI II scale implies to walk with two crutches plus both legs braces and no physical assistance throughout 10 m, which is more a therapeutic than community locomotion, not able to free the patient from the wheelchair (Thomas Jefferson University. Copyright 2004). Regarding another neurological consequences of having a chronic SCI, our patients showed mild-moderate levels of spasticity (mean Ashworth and Penn scores, 1.4821 ± 0.1372 and 1.5536 ± 0.1650 respectively) and pain (mean basal nociceptive and neuropathic VAS score = 0.3214 ± 0.1198 and 0.9107 ± 0.2474 respectively; mean VAS score during pain nociceptive and neuropathic crisis = 0.3214 ± 0.1198 and 0.9107 ± 0.2474 respectively). Finally, to ensure that only neurologically stable patients were recruited, a magnetic resonance scan and sensory and motor evoked potentials were performed, excluding from the present study those patients who, in the opinion of the radiologist and/or neurophysiologist, were worse than in previous studies. In the MRI images, the presence of syringomyelia was specifically addressed as an evolutionary complication in both traumatic and non-traumatic patients. Only 0.02% and 0.18% of our cases respectively presented it at the radiologist’s discretion. Only those syringomyelia that had not been modified in the last year were included.

The deficits derived from the autonomic nervous system damage were evident in the bladder (mean bladder ASIA Autonomic Standard Assessment score = 2.6250 ± 0.2016 over a maximum value of 6) and bowel (mean Bowel ASIA Autonomic Standard Assessment score = 2.5357 ± 0.2101) function of our patients, but although they suffered an average of less than 2 UTIs in the previous year, a higher rate than the general population (Flores-Mirelles et al., 2015; Foxman et al., 2000), no urinary or intestinal relevant complication was reported. With regard to the comorbidity conditions, the patients of our series showed non significative levels of fatigue (mean FSS scale = 2.3988 ± 0.1985) and only mild levels of anxiety (mean anxiety HAD score = 5.9464 ± 0.5568; mean anxiety EADG score = 2.7321 ± 0.3852) and depression (mean depression HAD score = 3.4107 ± 0.5082; mean depression EADG score = 2.0357 ± 0.3293). No statistically significant differences were found between traumatic and non-traumatic patients regarding to these parameters. No significant differences were found either between SCI patients and healthy controls with respect to age or sex distribution, neither in the clinical nor analytical variables studied except in two points: the comorbidity was significatively higher (mean SCI patients Charlson score = 2.7679 ± 0.1299 versus controls = 0.0909 ± 0.290; *p* < 0.001) and the One’s own health status perception significatively lower (mean SCI EQ5D-VAS score = 68.8929 ± 3.3782 versus controls = 84.975 ± 1.358; *p* < 0.001) between SCI patients and healthy controls, pointing to the intense impact that a chronic SCI, even the less complicated of them, has in a patient’s life.

### 2.2. Chronic SCI Patients Show Normal Monocyte Subset Distributions and Cell Counts but Abnormal TLR Expression

First, we studied the absolute number of circulating monocytes and their percentages in PBMCs from 56 patients with chronic SCI and 28 sex- and age-matched HCs. There were no statistically significant differences in the number of circulating monocytes between SCI subjects and HCs (Figure 1a). Next, we studied the distribution of the CD14^+high^CD16^−^, CD14^+high^CD16^+^ and CD14^+low^CD16^+^ monocyte subsets in both groups of subjects (Figure 1). There were no significant differences in the percentages of the CD14^+high^CD16^−^, CD14^+high^CD16^+^ and CD14^+low^CD16^+^ monocyte subsets in the circulating monocyte population between chronic SCI patients and HCs.

We also investigated the expression of TLR2, TLR4 and TLR9 in circulating monocytes from both groups of subjects (Figure 2). We found a significant decrease in the percentage of circulating monocytes that expressed TLR4 in patients with chronic SCI compared to the percentage observed in HCs. Concomitant with that change, we observed a significant increase in the percentage of monocytes that expressed TLR9 in SCI patients compared to the percentage observed in HCs. In contrast, we found no significant differences in the percentage of monocytes that expressed TLR2. We also analyzed TLR4, TLR9 and TLR2 expression in the different monocyte subsets from chronic SCI patients and HCs. We found a significant decrease in the percentages of the CD14^+high^CD16^−^ and CD14^+high^CD16^+^ monocyte subsets that expressed TLR4 in SCI patients compared to the percentages observed in HCs. In contrast, we observed significant increases in the percentages of CD14^+high^CD16^−^ and CD14^+high^CD16^+^ monocytes that expressed TLR9 in SCI patients compared to the percentages observed in HCs. No differences in the expression of TLR2 in the CD14^+high^CD16^−^, CD14^+high^CD16^+^, and CD14^+low^CD16^+^ monocyte subsets were observed between the two groups of subjects.

### 2.3. Chronic SCI Patients Exhibit TNF-α Overproduction by Monocytes and Increased Serum Levels of the Proinflammatory Cytokines TNF-α and IL-6

The intracellular expression of TNF-α, IL-1β, IL-6 and IL-10 was analyzed in monocytes from SCI patients and HCs after LPS stimulation (Figure 3). We found a significant increase in the percentage of monocytes that produced TNF-α after LPS stimulation in chronic SCI patients compared to the percentage observed in HCs. We found no significant differences in the percentages of monocytes that expressed IL-1β, IL-6, and IL-10 between the two groups of subjects.

Next, we studied the expression of these cytokines in the CD14^+high^CD16^−^, CD14^+high^CD16^+^ and CD14^+low^CD16^+^ monocyte subsets after LPS stimulation. We found that the CD14^+high^CD16^−^ monocyte subset from chronic SCI patients shows increased TNF-α expression compared to the expression level of the same subset in HCs. In contrast, in the three analyzed monocyte subsets, there were no significant differences in the percentages of monocytes that expressed IL-1β, IL-6, and IL-10 after LPS stimulation between the two groups of subjects.

We also measured the circulating levels of the proinflammatory cytokines TNF-α, IL-1β and IL-6 and the anti-inflammatory cytokine IL-10 (Figure 4). Subjects with chronic SCI had significantly higher levels of TNF-α and IL-6 than did HCs. Most chronic SCI patients presented higher TNF-α levels (70.83%) than did the top quartile of HCs.

Moreover, in chronic SCI patients, we found a significant direct correlation between serum TNF-α levels and LPS-induced TNF-α production in monocytes (r = 0.43, *p* < 0.05) and in the CD14^+high^CD16^−^ (r = 0.48, *p* < 0.05) and CD14^+low^CD16^+^ (r = 0.42, *p* < 0.05) monocyte subsets (Figure 5). We also found a significant negative correlation between serum TNF-α concentrations and TLR4 expression in the monocyte population (r = −0.31, *p* < 0.05) and in the CD14^+high^CD16^−^ (r = −0.31, *p* < 0.05) and CD14^+high^CD16^+^ monocyte subsets (r = −0.32, *p* < 0.05) but not in CD14^−^CD16^+^ monocytes (r = −0.2, *p* < 0.24). We found no significant correlations between serum TNF-α levels and TLR2 and TLR9 expression in monocytes from the two groups of subjects. In the HC group, there were no significant correlations among serum TNF-α levels, LPS-induced monocyte TNF-α expression, and TLR expression in circulating monocytes (data not shown).

### 2.4. Monocytes from Chronic SCI Patients Show Defective Phagocytic Effector Functions

We investigated phagocytosis and ROS production by monocytes after *E. coli* stimulation in chronic SCI patients and HCs (Figure 6). We found a significant decrease in the percentage of monocytes that phagocytosed *E. coli* in SCI patients compared to the percentage observed in HCs (Figure 6c). However, we found no significant differences in ROS production in monocytes from chronic SCI patients (Figure 6d).

### 2.5. Chronic SCI Patients Show Increased Levels of Circulating LBP, I-FABP and Zonulin

We also investigated the serum concentrations of LBP, I-FABP and zonulin in chronic SCI patients and HCs (Figure 4e–g). We found that chronic SCI patients show significantly increased LBP serum levels compared to those found in HCs. Specifically, 60.47% percent of chronic SCI patients presented higher LBP levels than did the top quartile of HCs. We also found that serum I-FABP and zonulin concentrations were significantly higher in chronic SCI patients than in HCs. Specifically, 72% and 66% percent of chronic SCI patients presented higher I-FABP and zonulin levels, respectively, than did the top quartile of HCs.

We also analyzed the different immune system parameters studied in the chronic SCI patient population stratified by the level of the spine lesion (from C1 to T5 versus from T6 to L6) (Figure 7a). We found that both groups of patients showed significantly higher TNF-α production and serum levels of TNF-α, LBP, I-FABP, and zonulin and lower TLR4 expression in monocytes than did HCs. There were no significant differences between the two groups of patients.

Finally, we classified chronic SCI patients into two groups according to their AIS scores: group 1, which covered A and B AIS scores, and group 2, which covered C, D and E AIS scores. We found a significant increase in the serum concentration of LBP in group 1 and group 2 compared to the concentration observed in HCs, as well as a significant decrease in group 2 compared to group 1 (Figure 7b). Monocyte TNF-α production as well as serum levels of I-FABP, zonulin and TNF-α were significantly elevated in both groups compared to the levels observed in HCs. We also found a significant decrease in the percentage of monocytes that expressed TLR4 in group 1 compared to the values observed for HCs.

## 3. Discussion

In this paper, we have demonstrated that chronic SCI patients without associated inflammatory and infectious diseases show functional impairment of circulating monocytes, with diminished TLR4 expression, increased LPS-induced TNF-α production and defective phagocytosis. These patients also show a systemic proinflammatory state characterized by enhanced serum TNF-α and IL-6 levels. Furthermore, augmented circulating levels of LBP, I-FABP, and zonulin are found in chronic SCI patients, indicating increased bacterial translocation and gut barrier damage.

The function of the immune system in patients with chronic SCI remains poorly defined. Contradictory results have been reported for T lymphocyte counts, subset distributions, and functions [21,33,34,35,36,37]. Monocytes are a cornerstone of the immune system that links innate and adaptive immunity and plays critical roles in the response to bacterial infections and in the induction and regulation of the inflammatory response [23]. Our findings show that chronic SCI patients have normal monocyte counts, which is consistent with a previous report [36]. Furthermore, reduced monocyte counts have been reported 4–5 months after SCI in acute patients [37]. Moreover, the distribution of the CD14^+high^CD16^−^, CD14^+high^CD16^+^, and CD14^+low^CD16^+^ circulating monocyte subsets is also normal in these patients.

Members of the TLR family play critical roles as regulators of innate and adaptive immune responses. Interestingly, the expression of TLR4 on monocytes from chronic SCI patients is diminished, and this observation can be explained by the observed reduction in the CD14^+high^CD16^−^ and CD14^+high^CD16^+^ subsets. However, chronic SCI patients show normal TLR2 expression on monocytes and in all three monocyte subsets. Alterations in monocyte TLR4 expression have been described in acute and chronic diseases. TLR4 overexpression on monocytes has been found in different noninfectious diseases, such as atrial fibrillation and major depression [38]. In contrast, sepsis patients show decreased TLR4 expression on monocytes that has been associated with worse outcomes and mortality [39,40]. We also found increased TLR9 expression on monocytes from chronic SCI patients, which could be explained by the enhanced expression found on the CD14^+high^CD16^−^ and CD14^+high^CD16^+^ monocyte subsets. TLR9, which binds bacterial DNA, is an intracellular molecule that is also found on the cell surface [10,41,42]. It has been proposed that the cell surface form of TLR9 binds bacterial DNA and that the ligand is then transferred from the cell surface to the intracellular compartment [42,43]. Increased TLR9 expression has been reported in chronic infectious and noninfectious inflammatory diseases, such as systemic lupus erythematosus, cutaneous leishmaniasis, chronic hepatitis B, and acute sepsis [44,45,46,47].

Monocytes can produce a plethora of cytokines that are essential for the adequate regulation of immune responses. In this study, we show a clear enhancement of monocyte TNF-α production, but the levels of IL-1, IL-6 and IL-10 were normal. The increased monocyte TNF-α expression observed in chronic SCI patients is linked to the CD14^+high^CD16^−^ subset. It is known that monocytes, of which a majority of cells are in the CD14^+high^CD16^−^ subset, secrete cytokines, including TNF-α, IL-1, IL-6 and IL-10, and release inflammatory mediators when stimulated with LPS [48]. In contrast, CD14^+high^CD16^+^ and CD14^+low^CD16^+^ monocytes from chronic SCI patients show normal TNF-α, IL-1, IL-6 and IL-10 production by monocytes after LPS stimulation. These CD14^+low^CD16^+^ monocytes mainly respond to nucleic acid stimulation [49]. Interestingly, decreased secretion of IP-10 by monocytes from chronic SCI patients has been described [36]. Thus, patients with chronic SCI show a specific proinflammatory pattern of altered monocyte cytokine production. This abnormal pattern of monocyte subset distribution has not been previously described in inflammatory diseases [48,50,51]. Furthermore, we found a marked defect in *E. coli* uptake but normal ROS generation in monocytes from patients with chronic SCI. These results indicate an impaired phagocytic ability but preserved monocyte intracellular microbicidal activity in these patients. Interestingly, monocyte ROS generation is preserved or even enhanced in septic patients, suggesting that oxidative metabolism and cytokine production are differentially regulated in monocytes from septic patients [52].

Increased serum levels of TNF-α and IL-6 have been associated with inflammatory diseases [53,54]. Consistent with previous studies, we found an increase in the inflammatory mediators TNF-α and IL-6 but normal IL-1β and IL-10 levels in chronic SCI patients [20,55]. Interestingly, serum TNF-α levels correlate with the production of this cytokine by LPS-stimulated monocytes from chronic SCI patients. The source of the increased IL-6 levels cannot be ascribed to monocytes since the production of this cytokine was normal. Several immune and nonimmune cells may produce IL-6, and its production can be induced by TNF-α [56].

Several mechanisms may be involved in the pathogenesis of the proinflammatory monocyte abnormalities found in chronic SCI patients, and our findings help to improve our understanding of this immune system alteration. To the best of our knowledge, this is the first report of elevated circulating LBP levels in chronic SCI patients. The hepatic synthesis of LBP is promoted by LPS, and LPS-LBP complexes bind to CD14 on the monocyte surface. LBP peaks in the plasma 2 to 3 days after transient bacteriemia or endotoxemia, and the levels remain increased up to 72 h later [30]. Indeed, in several clinical settings, plasma LBP seems to better reflect long-term exposure to bacteria and their endotoxins than endotoxin itself [32,57]. Increased serum LPS levels have been associated with higher circulating levels of TNF-α and IL-6 and increased TNF-α expression on monocytes [58]. Furthermore, increased LBP production has been associated with decreased expression of TLR4 on host immune cells [39]. Thus, the clinical setting described here differs from that of sepsis and septic patients, in which massive, acute LPS exposure promotes very high LBP concentrations that inhibit the LPS response by human monocytes and is therefore thought to be protective [59].

It was recently reported that experimental SCI is associated with increased intestinal permeability and bacterial translocation from the gut [60]. We investigated gut barrier damage in chronic SCI patients by quantifying I-FABP and zonulin, which are validated markers [61,62]. The increased serum I-FABP and zonulin levels observed in chronic SCI patients might reflect a loss of the integrity of the intestinal barrier. Increased intestinal permeability favors bacterial translocation [63]. Therefore, increased intestinal permeability in SCI patients may play a role in the observed increase in LBP levels. The urinary tract does not appear to be a relevant source of bacterial translocation in our chronic SCI patients since they had neither clinical manifestations of UTI nor positive urine cultures. Furthermore, the increased LBP levels cannot be explained by a clinical infection because the presence of recent acute or chronic infections was a patient exclusion criterion in our study.

In animal models, the immune disturbance found in acute SCI has been related to the involvement of the sympathetic autonomous nervous system and neuroimmune regulation [64]. However, in humans, the relevance of this impairment has not been established. Contradictory reports about the relationship between the level of injury and the impairment of NK cells have been made [6,33,34,35]. Our results show that gut barrier damage, bacterial translocation, monocyte disturbance, and systemic proinflammatory conditions in chronic SCI patients are independent of the thoracic level of the lesion.

The monocyte dysfunction and systemic proinflammatory cytokine pattern that are associated with increased bacterial translocation in chronic SCI patients provide clues to improve our understanding of the clinical complications that develop in these patients. Impaired monocyte phagocytic activity and defective TLR4 expression in the CD14^+high^CD16^−^ and CD14^+high^CD16^+^ monocyte subsets might be involved in common causes, predominantly septicemia, of death in the years following SCI [65,66]. Indeed, TLR4 plays a critical role in the clinical response to intraperitoneal *E. coli,* and TLR4 modulates the phagocytosis of bacteria by peritoneal macrophages [67]. Furthermore, increased bacterial translocation defined by enhanced LPS levels has also been associated with an increased prevalence of infections and mortality in different clinical settings, including those with augmented gut barrier damage and monocyte dysfunction [28,58].

Our findings demonstrate that patients with chronic SCI suffer an unexpected systemic inflammatory state, with severe disturbances of monocyte function and pathogenic compromise of the intestinal barrier, as summarized in Figure 8. Chronic SCI patients show a critical compromise of the intestinal barrier, as shown by the enhanced plasma levels of I-FABP and zonulin. This impairment favors a subsequent increase in bacterial translocation, as demonstrated by the augmented LBP levels found in chronic SCI patients. This bacterial pressure appears to be involved in maintaining monocyte overstimulation. Furthermore, monocytes from chronic SCI patients display defective phagocytic activity. These findings demonstrate that chronic SCI patients suffer not only a motor, sensory and/or autonomous nervous system disease but also a systemic inflammatory disease. These patients have severe comorbidities, such as premature coronary heart disease, metabolic syndrome and diabetes mellitus [10,68]. Activated monocytes and proinflammatory cytokines are clearly involved in the pathogenesis of accelerated atherogenesis and insulin resistance in different chronic inflammatory diseases [69,70]. Thus, the immune disturbances found in chronic SCI patients may also favor the appearance of these high-morbidity diseases. Taken together, our findings suggest that we should consider noncomplicated chronic SCI as a systemic inflammatory disease with monocyte dysfunction and increased bacterial translocation across the intestinal barrier.

One limitation to this study might be the limited number of participants. In addition, we have not carried out a longitudinal study of the patients. Moreover, the immune system is a complex system that involves many populations and molecules that must be analyzed together. Therefore, it would be interesting to analyze other compartments of the immune system as well as their functional interaction between them

Our findings provide an innovative understanding of chronic SCI that will support the development of new strategies for preventing the severe comorbidities that these patients suffer. Future translational and longitudinal clinical studies must define the potential biomarker value of immune system and/or bacteria-dependent host parameters for the development of infectious and noninfectious complications. Furthermore, these results support the need to investigate new immunomodulatory and microbiological strategies in patients with chronic SCI. These immune disturbances and the associated bacterial translocation must be analyzed in the whole population of patients, including those with inflammatory or septic complications.

## 4. Materials and Methods

### 4.1. Study Protocol

In this prospective study, we included 56 chronic SCI patients using the following inclusion criteria: (1) ≥18 years of age; (2) a history of SCI, with at least 1 year of SCI at any level; (3) SCI with any severity, including grades A to E classified with the American Spinal Injury Association (ASIA) Impairment Scale (AIS). A physiatrist board-certified in SCI medicine evaluated the subjects’ injuries according to the International Standards for Neurologic Classification of Spinal Cord Injury [71,72]. Potential subjects were excluded if they had (1) a concurrent infection complication, such as a urinary tract infection (UTI) or a respiratory infection with positive urine culture in the last three months; (2) chronic bacterial or viral infection; (3) pressure ulcers in the last 12 months; (4) received steroids or immunomodulatory drugs in the last three months; (5) an autoimmune disease; (6) a severe cardiovascular disease; (7) a hematopoietic, lung, hepatic or renal disorder; (8) an endocrine or metabolic disease, including diabetes mellitus; (9) a history of malignancy; (10) immunodeficiency and malnutrition; (11) pregnancy or lactation; and (12) psychiatric disorders. The patients were studied in parallel with 28 sex- and age-matched HCs.

The experimental protocol included a detailed clinical assessment that encompassed several clinical parameters—some of them directly related to the SCI but others needed for the correct interpretation of immunological data—and a blood sample for the determination of routine hematological and biochemical parameters along with the immunological parameters detailed below. Clinical data from the SCI subjects were obtained during a routine medical examination in an outpatient clinic in the Physical Medicine and Rehabilitation Department, and included: (1) baseline demographic characteristics; (2) time from and mechanism of initial injury; (3) neurologic injury level and severity; (4) tonic and phasic spasticity; (5) presence, type and severity of pain; (6) medical history of infections and other symptoms evocative of some chronic SCI complication; (7) comorbid conditions; (8) concurrent medications; (9) fatigue; (10) depression and anxiety levels; (11) level of independence in daily living activities; (12) and a measure of one’s own quality of life and health status perception.

This study was approved by the institutional and regional clinical ethics committee. Written informed consent was obtained from all subjects before study enrollment.

Clinical data from the SCI patients were obtained during a routine medical examination at an outpatient clinic in the Physical Medicine and Rehabilitation Department and included the following. This study was approved by the institutional and regional clinical ethics committee. Written informed consent was obtained from all subjects before study enrollment.

Blood samples were drawn from all subjects via standard venipuncture using established aseptic technique. Samples were obtained from the chronic SCI patients at the time of the clinical evaluation in the outpatient clinic area. Serum samples from 56 subjects with chronic SCI and 28 uninjured subjects were included for analysis. After collection, the samples were centrifuged, and the serum was isolated, aliquoted, and stored at −80 °C until analysis.

### 4.2. Isolation of Peripheral Blood Mononuclear Cells

Peripheral blood mononuclear cells (PBMCs) were separated by Ficoll-Hypaque (Lymphoprep^TM^, Axis-Shield, Oslo, Norway) gradient centrifugation. The cells were then resuspended in RPMI 1640 (BioWhittaker Products, Verviers, Belgium) supplemented with 10% heat-inactivated fetal calf serum, 25 mM HEPES (BioWhittaker Products) and 1% penicillin-streptomycin (BioWhittaker Products). Cell enumeration was performed by conventional light microscopy using a Neubauer chamber following the criteria for trypan blue dead cell exclusion.

### 4.3. Immunophenotype Studies

The proportions of monocyte subsets were determined in fresh PBMCs by ten-color polychromatic flow cytometry in a FACSAria cytometer using FACSDiva software (Becton Dickinson, NJ, USA). One million PBMCs were incubated with a combination of the following monoclonal antibodies (MoAbs): CX3CR1-FITC, Slan-PE, HLA-DR-PerCP, CCR2-PerCP-Cy5.5, CD11c-PE-CY7, CD3/CD56/CD19-APC, CD62L-APC-A780, CD16-PB, Aqua-QD565, and CD14-QD655. For these procedures, CD14-QD655, CD16-Alexa405, HLA-DR-PerCP, CD3-APC, CD19-APC, CD56-APC, CCR2-PerCP-Cy5.5, and CD62L-Alexa780 were obtained from Becton Dickinson, CX3CR1-FITC and CD11C-PE-CY7 were obtained from e-Biosciences (e-Biosciences, San Diego, CA, USA), SLAN-PE was obtained from Miltenyi (Miltenyi, Bergisch Gladbach, Germany) and Aqua-QD565 was obtained from Invitrogen (Invitrogen, Carlsbad, CA, USA). 

The expression of TLRs on monocyte subsets was determined in fresh PBMCs by five-color polychromatic flow cytometry in a FACSAria cytometer using FACSDiva software (Becton Dickinson). Fresh PBMCs were labeled with CD14-QD655, CD16-Alexa405, TLR2-FITC, TLR4-APC and TLR9-PE (Becton Dickinson) MoAbs.

For all samples, once the MoAbs were added, the cells were incubated for 20 min at 4 °C in the dark. After that time, the cells were washed in phosphate-buffered saline (PBS) to eliminate excess antibody, and 100 µL of PBS was added for subsequent acquisition by flow cytometry. Analyses were carried out using FlowJo software (TreeStar Inc., Ashland, OR, USA).

### 4.4. Intracellular Cytokines

To analyze the production of cytokines by PBMCs, fresh PBMCs were cultured in ultralow attachment plates (Corning Incorporated, Baltimore, MD, USA) (1 mL of cells at 10^6^ cells/mL) and incubated for 4 h at 37 °C with 5% CO_2_. PBMC stimulation was performed by adding LPS (5 µg/mL, Sigma-Aldrich Chemistry, Madrid, Spain) and monensin (50 µg/mL, Sigma). Next, the cells were labeled with CD14-PerCP and CD16-Alexa647 (Becton Dickinson) MoAbs and the vital dye Aqua-QD565. For intracytoplasmic staining, the cells were fixed and permeabilized (Fix and Perm, Caltag Laboratories, Burlingame, CA, USA), and cytokines were stained with IL-1β-FITC, IL-10-PE, IL-6-V505, and TNF-α-Alexa700 (Becton Dickinson) MoAbs.

### 4.5. Flow Cytometry Studies of Oxidation and Phagocytic Activity

The phagocytic function of circulating monocytes was determined by the intake of *Escherichia coli* (Phagotest; Becton Dickinson) and the quantification of the oxidative burst activity of monocytes in heparinized human whole blood was determined by the PHAGOBURST test (Becton Dickinson).

### 4.6. Quantification of Serum Cytokines Using Luminex

To study the concentrations of cytokines in the serum, the Milliplex MAP Kit (MERCK, Darmstadt, Germany) was employed using the protocol recommended by MERCK. The plate was read in a Luminex MAGPIX with xPONENT software (Luminex Corporation, Northbrook, IL, USA).

### 4.7. Study of Damage to the Intestinal Barrier

To study intestinal barrier damage, an analysis of I-FABP and zonulin concentrations in the serum was performed by ELISA. I-FABP was purchased from Hycult Biotech (Hycult Biotech, Wayne, PA, USA), and zonulin was purchased from R&D Systems (R&D Systems, Minneapolis, MN, USA). The plate was read in an iMark Microplate Reader at 450 nm with Microplate Manager Software (Thermo Fisher Scientific, Frederick, MD, USA).

### 4.8. Study of the Acute Phase Response

To study the acute phase response, we determined the concentration of LBP in the serum by ELISA (Abnova, Taipei, Taiwan). The plate was read in an iMark Microplate Reader at 450 nm with Microplate Manager Software (Thermo Fisher Scientific).

### 4.9. Statistical Analysis

Comparisons between patients and HCs were performed using the nonparametric Mann–Whitney U test. Associations between variables were assessed with the Spearman’s rank correlation coefficient by simple linear regression analyses. All calculations were performed using the Statistical Package for the Social Sciences (SPSS, version 22.0, Chicago, IL, USA). Significance was set at *p* < 0.05.

## Figures and Tables

**Figure 1 ijms-22-00744-f001:**
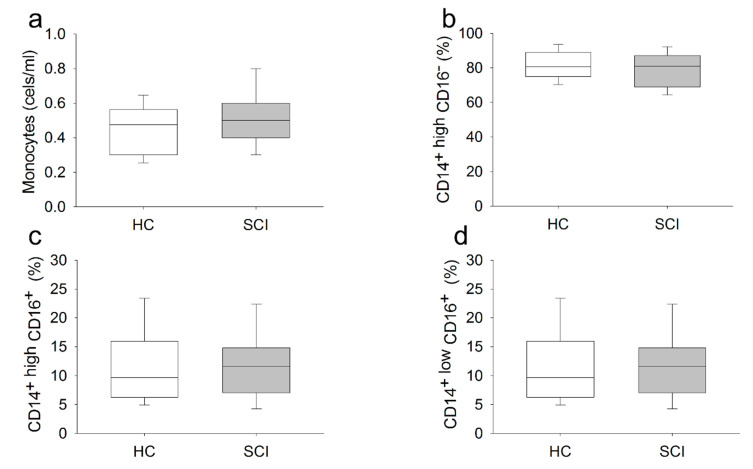
Absolute number of circulating monocytes and the distribution of monocyte subsets in chronic SCI patients. Absolute number (cells/μL) of circulating monocytes: (**a**) and the percentages of the CD14^+high^CD16^−^; (**b**) CD14^+high^CD16^+^; (**c**) and CD14^+low^CD16^+^; (**d**) monocyte subsets in chronic SCI patients (gray box) and healthy controls (white box).

**Figure 2 ijms-22-00744-f002:**
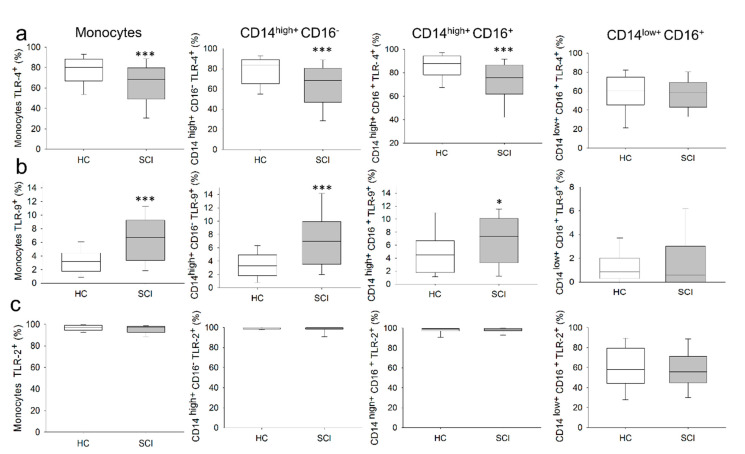
Expression of TLR4, TLR9 and TLR2 in circulating monocyte subsets from chronic SCI patients. Percentages of TLR4: (**a**) TLR9; (**b**) and TLR2; (**c**) in circulating monocytes and the CD14^+high^CD16^−^, CD14^+high^CD16^+^ and CD14^+low^CD16^+^ monocyte subsets from chronic SCI patients (gray box) and healthy controls (white box). * Significant difference between patients and healthy controls (*p* < 0.05). *** Significant difference between patients and healthy controls (*p* < 0.001).

**Figure 3 ijms-22-00744-f003:**
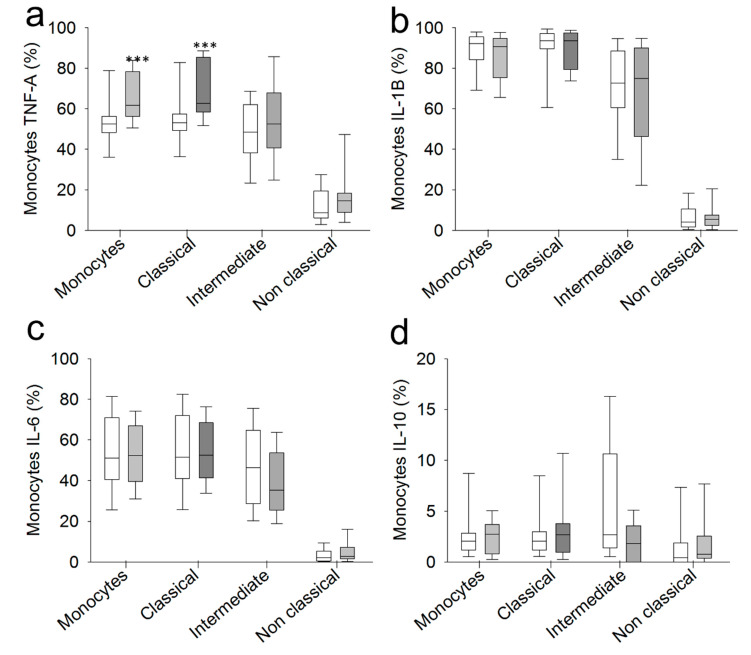
Cytokine production by circulating monocytes from chronic SCI patients. Percentages of monocytes producing TNF-α: (**a**) IL-1β; (**b**) IL-6; (**c**) and IL-10 (**d**) among circulating monocytes and the CD14^+high^CD16^−^, CD14^+high^CD16^+^ and CD14^+low^CD16^+^ monocyte subsets from chronic SCI patients (gray box) and healthy controls (white box). *** Significant difference between patients and healthy controls (*p* < 0.001).

**Figure 4 ijms-22-00744-f004:**
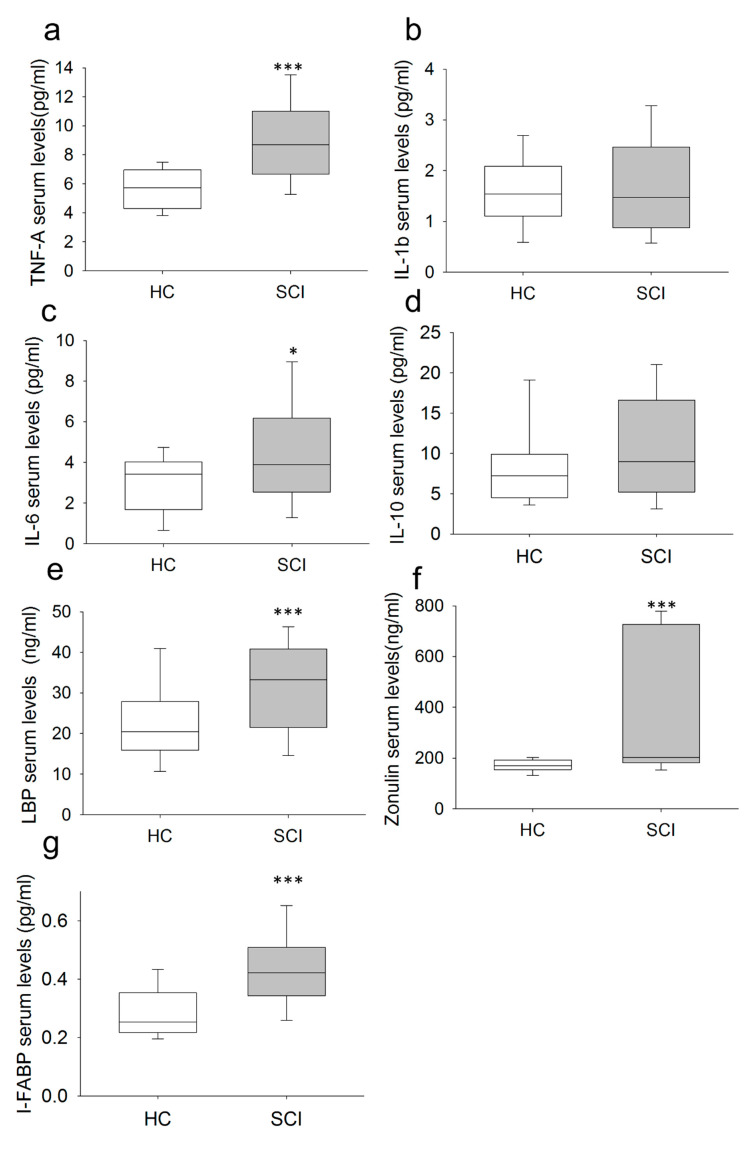
Determination of serum soluble mediators in chronic SCI patients. Serum concentrations of TNF-α: (panel (**a**)) IL-1β; (**b**) IL-6; (**c**) IL-10; (**d**) LBP; (**e**) zonulin; (**f**) and I-FABP; (**g**) in chronic SCI patients (gray box) and healthy controls (white box) are shown. * Significant difference between patients and healthy controls (*p* < 0.05). *** Significant difference between patients and healthy controls (*p* < 0.001).

**Figure 5 ijms-22-00744-f005:**
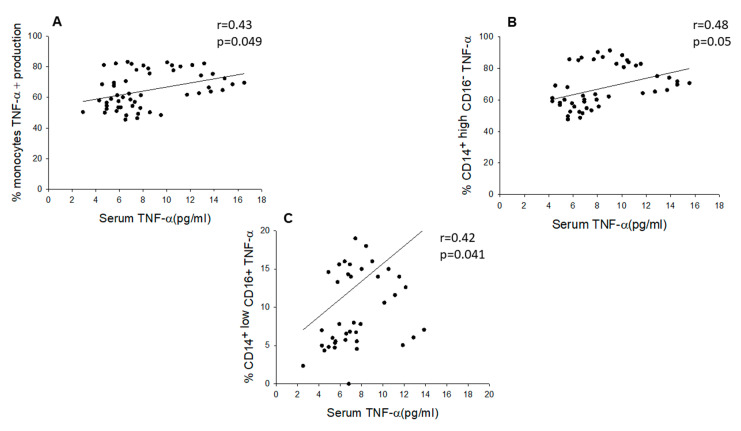
Serum TNF-α levels directly correlates with LPS-induced TNF-α production in monocytes their CD14+highCD16- and CD14+lowCD16+ subsets. Correlations between serum TNF-α levels and monocytes (**A**), CD14+highCD16- (**B**) and CD14+lowCD16+ (**C**) subsets are shown.

**Figure 6 ijms-22-00744-f006:**
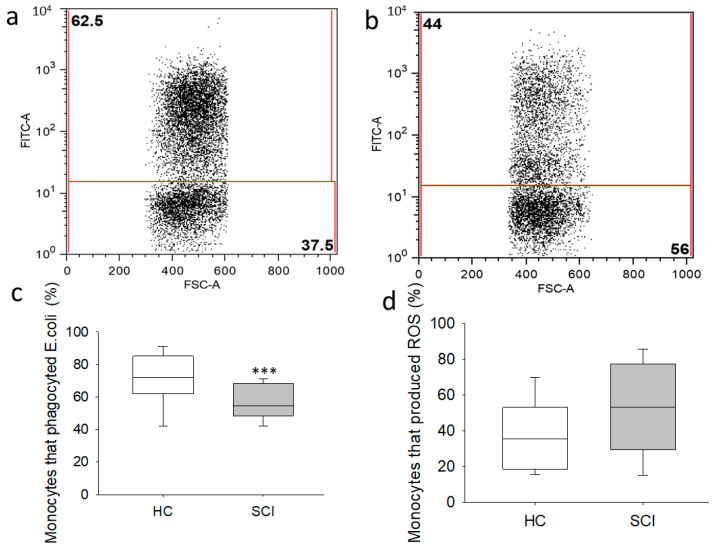
Phagocytosis and ROS production by circulating monocytes from chronic SCI patients. Representative analysis of monocytes that phagocytosed *E. coli* in a healthy control: (**a**) and a chronic SCI patient; (**b**). The percentages of peripheral blood monocytes that phagocytosed *E. coli;* (**c**) and that produced ROS; (**d**) in chronic SCI patients (gray box) and healthy controls (white box) are shown. *** Significant difference between patients and healthy controls (*p* < 0.001).

**Figure 7 ijms-22-00744-f007:**
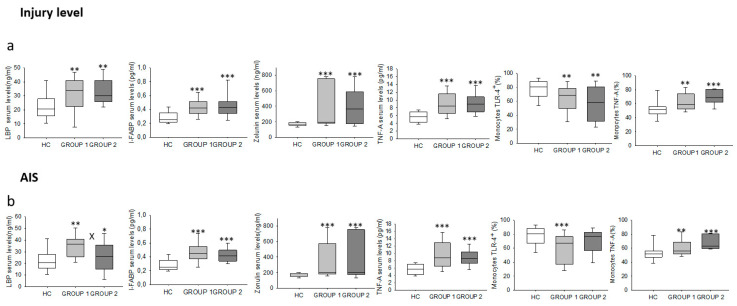
Clinical stratification of chronic SCI patients. Different immune system parameters (LBP, I-FABP, zonulin, serum TNF-α, TLR4 and TNF-α-producing monocytes) in chronic SCI patients (gray box) and healthy controls (white box) were analyzed. Patients were stratified by the level of the spine lesion: (**a**), from C1 to T5 (Group 1) versus from T6 to L6 (Group 2), and by the AIS score; (**b**), as follows: group 1 covers A and B AIS scores, and group 2 covers C, D and E AIS scores. * Significant difference between patients and healthy controls (*p* < 0.05). ** Significant difference between patients and healthy controls (*p* < 0.01). *** Significant difference between patients and healthy controls (*p* < 0.001). X Significant difference between both groups of patients (*p* < 0.01).

**Figure 8 ijms-22-00744-f008:**
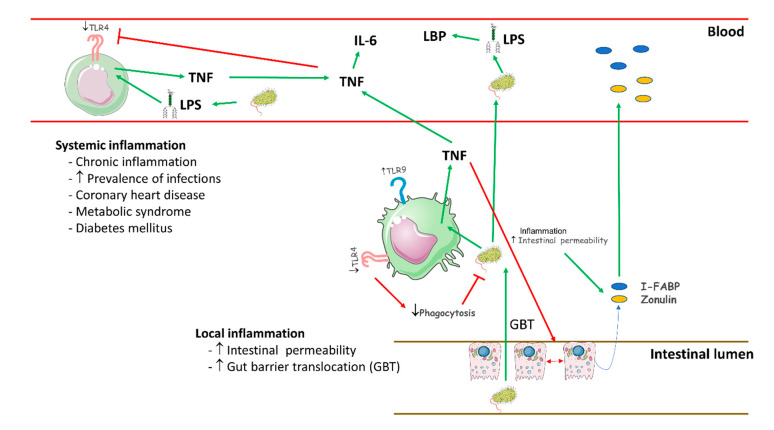
Schematic representation of pathology in chronic SCI patients.

**Table 1 ijms-22-00744-t001:** Demographic, clinical and biological data of the patients and healthy controls.

Variables	Healthy Controls	Chronic SCI	AIS Group 1	AIS Group 2	Injury Level Group 1	Injury Level Group 2
	(*n* = 28)	Patients (*n* = 55)	(A–B) (*n* = 30)	(C–D) (*n* = 25)	(C1–T6) (*n* = 35)	(T7–L6) (*n* = 20)

Age (years)	25.03 ± 2.86	26.92 ± 12.87	24.96 ± 12.90	30.65 ± 13.30	23.28 ± 13.64	27.21 ± 12.69
Sex (men%/women%)	43.24/56.76	68.00/32.00	76.67/23.33	60.86/39.14	74.28/25.71	57.89/42.10
Time of injury (years)		12.00 ± 9.22	10.7 ± 9.15	12.91 ± 9.61	10.74 ± 9.69	14.23 ± 8.81
AIS (%)						
A		34.00	63.33		48.57	10.52
B		21.43	36.67		14.28	26.31
C		19.64		45.83	8.57	36.84
D		25.00		54.17	28.57	21.05
Injury level (%)						
C1–C4		23.21	20.00	25.00	34.28	
C5–C8		19.64	20.00	16.67	28.57	
T1–T6		23.22	33.33	12.50	37.14	
T7–T12		19.64	13.33	29.17		57.89
L1–L6		14.29	13.33	16.67		42.10

AIS: American Spinal Injury Association (ASIA) Impairment Scale; SCI: Spinal Cord Injury.

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
