# Peer review of "Systemic Inflammation and the Breakdown of Intestinal Homeostasis Are Key Events in Chronic Spinal Cord Injury Patients"

_ijms, 2021, doi:10.3390/ijms22020744_

Round 1

Reviewer 1 Report

You mention a significant correlation between serum TNF-alpha levels and LPS-induced TNF-alpha in monocytes levels is significant.  Please do show this figure as a correlation graph with the R number.

Figure D of ROS production in monocytes from patients and controls. It appears to me that the ROS production in SCI patients appears significantly higher. Although you state that their is no difference.  Can you check this please? It may be that their is one ''outlier''.

Figures 5 and 4 legends are mixed up.

Author Response

The manuscript has been thoroughly revised to address the recommendations made by the reviewer. We have addressed all the changes suggested by the reviewer and have included a new figure with the data requested.

The answers to the reviewers’ concerns and suggestions are given in the following point by point list:

1) We thank the reviewer for his/her comments on our work. In agreement with the reviewer’s comment, we provide a new figure (Figure 5) in the original manuscript (line 249), including images showing the correlation between serum TNF-alpha levels and LPS-induced TNF-alpha in monocytes and their CD14+highCD16- and CD14+lowCD16+ subsets (lines 259-1262). We also show r and p value. Accordingly, the numbering of the following figures has been modified.

2) As the reviewer points out, it appears that the ROS production in SCI patients appears slightly higher. To verify this, we have repeated the statistical analysis and indeed there are no significant differences between both groups (p= 0.268).

3) According to the request made by the reviewer, the legends have been exchanged (lines 242-246).

Reviewer 2 Report

This is a well conducted study that investigated the subset distribution and function of circulating monocytes,25 proinflammatory cytokine levels, gut barrier damage and bacterial translocation in chronic spinal cord injury (SCI) patients without associated inflammatory and infectious diseases. Functional impairment of circulating monocytes, with diminished TLR4 expression, increased LPS-induced TNF-α production and defective phagocytosis was found. A systemic proinflammatory state was also noted in the study materal. Furthermore, augmented circulating levels of LBP, I-FABP and zonulin are found in chronic SCI
232 patients, indicating increased bacterial translocation and gut barrier damage.

The provided information is interesting but the clinical perspective is poorly addressed. It is good to know how many were operated, how many times, what kind of surgery was done, what was the initial ASIA score and if it improved over time. Also chronic spinal cord injury patients frequently have clinical and radiological signs of posttraumatic myelopathy and syringomyelia. In how many patients was this entity detected, how it was managed and how it correlated with systemic inflammation and breakdown of intestinal homeostasis? Detailed radiological data are important for this purpose. Moreover, besides the control group, if possible, it would be good to have baseline data from the study itself to compare with. Lastly, a limitations paragraph is missing.

Author Response

The manuscript has been thoroughly revised to address the recommendations made by the reviewer.

The answers to the reviewers’ concerns and suggestions are given in the following point by point list:

1) The authors thank the reviewer´s comments, especially regarding the need for spinal surgery and the presence of syringomyelia as an evolutionary complication. Both aspects have been included in the new version of the article (lines 105-179).

2) We have finally decided not to include the comparison between the level and degree of severity of acute spinal cord injury compared to the current period, because our population is made up of chronic patients with an average evolution of more than a decade and both traumatic and non-traumatic, therefore heterogeneous. We considered that the best way to homogenize them was to focus on the absence of neurological changes in the exploration, evoked potentials and MRI image

A detailed description of the clinical profile of the patients included in the study has been included in the text  (lines 476-490).

3) We have also included the limitations of the study (lines 453-457): “One limitation to this study might be the limited number of participants. In addition, we have not carried out a longitudinal study of the patients. Moreover, the immune system is a complex system that involves many populations and molecules that must be analyzed together. Therefore, it would be interesting to analyze other compartments of the immune system as well as their functional interaction between them”.